# Dynamic ambulance relocation: a scoping review

Julia Becker,[1] Lisa Kurland,[2,3] Erik Höglund,[2,4] Karin Hugelius  [2,4]

[1]Institute for Disaster and Emergency Management, Berlin, Germany
[2]Örebro Univeristy, Faculty of Medicine and Health, Orebro, Sweden
[3]Örebro University Hospital, Orebro, Sweden
[4]Ambulance Department, Örebro Country Council, Örebro, Sweden

**Correspondence to**
Dr Karin Hugelius;
Karin.hugelius@oru.se

## ABSTRACT

**Objectives** Dynamic ambulance relocation means that the operators at a dispatch centre place an ambulance in a temporary location, with the goal of optimising coverage and response times in future medical emergencies. This study aimed to scope the current research on dynamic ambulance relocation.

**Design** A scoping review was conducted using a structured search in PubMed, Scopus and Web of Science. In total, 21 papers were included.

**Results** Most papers described research with experimental designs involving the use of mathematical models to calculate the optimal use and temporary relocations of ambulances. The models relied on several variables, including distances, locations of hospitals, demographic-geological data, estimation of new emergencies, emergency medical services (EMSs) working hours and other data. Some studies used historic ambulance dispatching data to develop models. Only one study reported a prospective, real-time evaluation of the models and the development of technical systems. No study reported on either positive or negative patient outcomes or real-life chain effects from the dynamic relocation of ambulances.

**Conclusions** Current knowledge on dynamic relocation of ambulances is dominated by mathematical and technical support data that have calculated optimal locations of ambulance services based on response times and not patient outcomes. Conversely, knowledge of how patient outcomes and the working environment are affected by dynamic ambulance dispatching is lacking. This review has highlighted several gaps in the scientific coverage of the topic. The primary concern is the lack of studies reporting on patient outcomes, and the limited knowledge regarding several key factors, including the optimal use of ambulances in rural areas, turnaround times, domino effects and aspects of working environment for EMS personnel. Therefore, addressing these knowledge gaps is important in future studies.

## BACKGROUND

Emergency medical services (EMSs) are an essential part of the emergency care system. An increasing number of emergency calls and EMS use, and the resultant influx into the emergency department, are driving the need for optimal use of existing resources.[1] For this reason, considerable research has been devoted to optimising the use of the EMS from a dispatching and logistical perspective.[2]

## STRENGTHS AND LIMITATIONS OF THIS STUDY

⇒ A broad, structured search strategy including several pilot searches, was performed in scientific databases and included papers with both medical and technical focus.
⇒ A quality appraisal of included papers was added to the original scoping review method.
⇒ A limitation was that the systematic search was conducted in three databases and included a limited number of papers reporting on dynamic relocation of ambulances from non-experimental settings and rural areas.
⇒ Another limitation was the exclusion of editorials, non-academic papers and publications other than those published in English.

Dispatching ambulances is a complex task, as dispatch decisions include both strategic and operational considerations, such as the number and location of fixed ambulance stations and how the available ambulances should be used to optimise ambulance coverage.[2] The responsiveness of the EMS depends on both stable factors, such as the location of a hospital, and factors that are constantly changing, such as traffic situations or weather conditions.[3] Two main models have been identified for operational decision-making on how best to use available resources and which ambulance to assign to a specific emergency call. The first model is the static location model (ie, the ambulance is dispatched to a specific/given ambulance station and it returns to its designated ambulance station after a completed mission). The second model is dynamic relocation,[2,4] also termed dynamic allocation or dynamic ambulance deployment. Dynamic relocation of ambulance vehicles means that the operator at the dispatch centre directs the ambulance to a temporary location (ie, a location other than the regular, static ambulance station) with the purpose of reducing the response time. In this review, the term dynamic ambulance relocation was used to describe this kind of flexible repositioning of available ambulance resources. The use of dynamic

BMJ

ambulance relocation in Sweden varies, from not used at all to over 20 relocation missions per 1000 inhabitants.[5] In the Örebro County Council (Sweden), the number of dynamic ambulance relocation missions increased from 2% in 2021 to over 5% in 2022.[6]

Many studies have focused on technical solutions or relied on artificial intelligence (AI) to support ambulance dispatch operators in allocating the ambulance to the 'best' location.[7] Most of the focus has been on time. Prehospital time is a crucial factor for the survival of critically ill or injured patients.[8 9] In particular, response time (eg, the time from dispatch until the ambulance is at the patient's side)[10] and turnaround time (eg, the time from being dispatched until the ambulance is ready for new assignments) seems to be critical.[11] However, little is still known about the dispatch model and prehospital time, and no consensus has been reached regarding the most effective model for ambulance dispatching. The increased burden on the EMS system and the considerable variation in the use of dynamic ambulance relocation have created a need to summarise the current scientific knowledge on dynamic ambulance relocation and to identify knowledge gaps in the field. Therefore, the review aims to scope the current research on dynamic ambulance relocation.

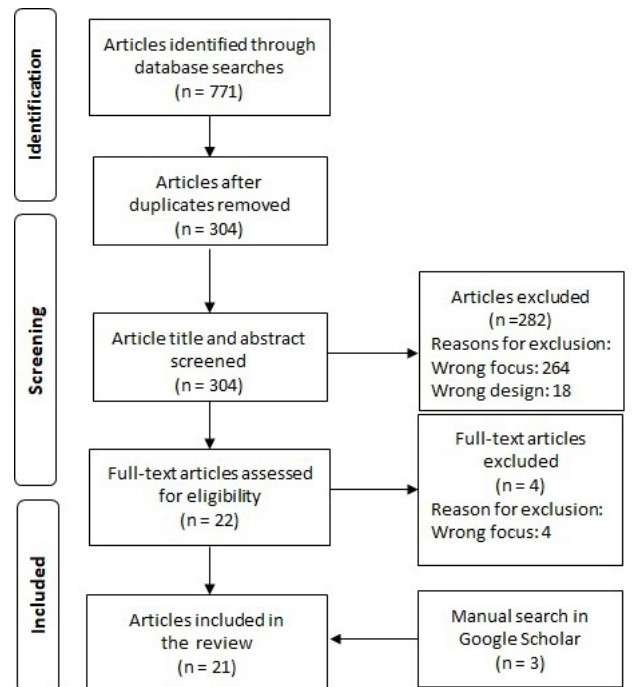

**Figure 1** Overview of the selection process and reasons for exclusion.

## METHODS

A scoping review was conducted using the Arksey and O'Malley methodology.[12] The Preferred Reporting Items for Systematic Reviews and Meta-Analyses (PRISMA) Scoping Reviews protocol was followed and used to report the study.

### Stage 1: identifying the research question

The research question for this study was 'What is known about dynamic ambulance allocations?'

### Stage 2: identifying relevant studies (search strategy)

Structured searches were performed in the PubMed database, Scopus and Web of Science by one author (KH) and two academic librarians using both MeSH terms and keywords, as well as truncation and Boolean operators (see online supplemental material 1). Pilot searches were made in December 2022 and repeated in July 2023. The final search was made in October 2023. The searches were limited to papers published in English between 2012 and 23 October 2023. In addition, a manual search in Google Scholar, using free text search with the terms 'dynamic ambulance' and 'ambulance relocation', sorted by 'best match' was performed in September and October 2023 by two authors (EH and JB) individually, which retrieved three additional papers in addition to the structured search.

### Stage 3: study selection

The study selection process was conducted using Covidence software (Covidence systematic review software, Veritas Health Innovation, Melbourne, Australia). All

papers identified in the literature search (n=771) were exported to the Covidence systematic review software, where duplications were sought and removed. Thereafter, all records were screened by the first and second authors (JB and KH) by reading the title and abstract. For inclusion, an original research paper had to report on dynamic ambulance allocation for road ambulances and be published between 2012 and 2023. Both qualitative and quantitative studies were included. Papers reporting on air ambulance services, fixed locations of ambulance resources (such as the optimal location of an ambulance station), review studies, case studies or editorial texts were excluded. The full-text review assessed the entire paper and its eligibility for the study. In this stage of the selection process, the reason for exclusion was documented (see figure 1). One disagreement occurred during the selection process, and it was solved by involving the second author (LK) in the decision.

In addition to the original scoping review methodology, a quality appraisal was performed,[13] using the Critical Appraisal Skills Programme checklist for cohort studies.[14] The quality appraisal was performed by the first author (JB) and validated by the fourth author (KH) (see online supplemental material 2).

### Stage 4: charting the data

All included studies were reviewed, and the following information was extracted: author(s) and year of publication, study location, design, aim, focus/aim of the study, setting (urban and/or rural areas), type of modelling, variables used, main findings and quality appraisal.

## Stage 5: collating, summarising and reporting the results

The results were synthesised and reported according to three domains: (1) study contexts and designs, (2) outcome variables and variables used in predictions of optimal ambulance dispatching and/or relocation and (3) outcomes from dynamic ambulance relocation.

## Patient and public involvement statement

No patients were included in the study.

## RESULTS

In total, 21 original papers were included in this scoping review. The included articles were published between 2012 and 2022, and the studies were conducted in Austria (n=1), Brazil (n=1), Canada (n=1), China (n=1), Colombia (n=1), Germany (n=1), Indonesia (n=1), Iran (n=2), Portugal (n=1), Singapore (n=3), Slovakia (n=1), Spain (n=1), Switzerland (n=1), the Netherlands (n=3), Turkey (n=1) and the USA (n=1). The studies were assessed as medium (n=16) or high (n=5) in quality (see online supplemental material 2).

## Study contexts and designs

The studies were most often conducted in urban areas, including metropolitan cities. Few studies were conducted in rural areas (see online supplemental material 3). One study[15] was only based on emergency calls for critical patients.

All studies were based on mathematical simulation models, with a large variation of outcome variables. Most studies (n=12) used an experimental design and simulations. Of these, most (n=8) simulations were based on retrospective EMS data and medical emergencies within a specified region (see table 1 and online supplemental material 3). Seven studies used experimental design (theoretical applications with no simulation) and two studies reported a prospective, real-time evaluation of the models and technical systems (see online supplemental material 3). Two studies evaluated their developed models in real-time evaluations, one in both urban and rural areas[16] and one in an urban context.[17]

## Outcome variables and variables used in predictions of optimal ambulance dispatching and/or relocation

Most studies (n=17) used response time as the primary outcome variable for the models developed and/or tested. The information used to develop the mathematical models and/or computer-based programmes was collected from retrospective data from real EMS activities (n=6), geographical and demographic data from national databases or Google Earth (n=6), estimations made by the research teams (n=3) or a combination of these sources within the same study (n=3) (see online supplemental material 3). Some studies did not specify the sources for their variable data (n=10). All studies included three or more variables in their models.

## Outcomes from dynamic ambulance relocation

No study reported or discussed medical outcomes or clinical effects of dynamic relocation of ambulances. Four studies reported decreased response times when using dynamic relocation compared with ordinary dispatching and when using ordinary ambulance stations. Reported time saved was 130–160 s,[18] 44 s,[19] 54 s[20] and undefined saving of time [71]. In one study, the average percentage of calls reached within 9 min was improved with 43% up to 55% in different simulations.[18] However, the same study showed an increase in total travelled distance by 7.9%–31.7%.[18]

No study reported on changes in response time or turnaround time in rural areas. Chain effects on other emergency calls due to dynamic ambulance relocations were reported.[16] One study reported that dynamic ambulance relocation reduced the total working time of all ambulances by about 9% per shift.[17]

## DISCUSSION

This review showed that most studies on dynamic ambulance relocation were based on experimental study designs and presented mathematical models in which three or more variables were used to calculate dynamic relocation of an ambulance with respect to response time. Most of the variables used in the models were fixed, and few studies evaluated the models or programmes in prospective, real-life settings. Most studies were conducted in rural areas. No study reported on patient outcomes attained using dynamic ambulance relocation.

Ambulance dispatching and the optimal location of ambulances are important but challenging questions to analyse. Most variables used in the models presented were static, gathered from GIS systems or similar techniques. New techniques, such as AI, may offer additional tools for decision-making and dispatching of ambulances in the future.[21] However, dispatchers also use other variables, such as medical priority, type of medical care required, clinical judgement and their own professional experience, when making decisions on ambulance dispatching.[22] Therefore, an important consideration is to identify how fixed variables can be incorporated into decision-making systems, while also recognising the limitations of these tools.

No consensus has yet been reached on how to assess, define or measure the quality of care within the prehospital setting[23]; however, prehospital response times have long been a core component in these discussions.[10] Moreover, response times are often available for research and are (therefore) used as a proxy for outcome, which is also reflected in the results as most of the studies included in this review used response time as the primary outcome. However, trauma has traditionally been used as a role model for time-critical medical conditions, as prolonged response time is associated with increased mortality in trauma victims,[24] especially in rural areas.[11] Studies suggest that prolonged prehospital time may have

**Table 1** Summary of included studies

| | | No of studies*<br>N=21 |
|---|---|---|
| Study design | Experimental design and simulations | 12 |
| | Experimental design with no simulation/testing | 7 |
| | Experimental design with real-time evaluation of models | 2 |
| Data used | Retrospective data from EMS activities | 4 |
| | Geographic and demographic data from national databases or Google Earth | 8 |
| | Estimations made by the research teams | 3 |
| | Combinations of estimations and official data | 3 |
| | No information given | 3 |
| Study context | Urban areas | 12 |
| | Rural areas | 2 |
| | Both urban and rural areas | 4 |
| | No information given | 2 |
| Outcome variable | Response time | 17 |
| | Turnaround time | 3 |
| | Workload on each ambulance | 2 |
| | Availability for new calls | 2 |
| | Perceived feasibility of software | 1 |
| | Travel distance per ambulance | 1 |
| Variables used in the models | Travel times and road network | 14 |
| | Current location of ambulances | 13 |
| | No of available ambulances | 9 |
| | Location of hospitals | 6 |
| | Population living in a defined geographical area | 6 |
| | Time of day | 4 |
| | Response time | 7 |
| | Expected occurrence of medical emergencies within a specified area or timeframe | 5 |
| | Type of emergency or priority of the call | 3 |
| | Turnaround time | 2 |
| | Time at scene | 2 |
| | Uncovered emergencies | 1 |
| | Economical costs | 1 |
| | EMS working hours | 4 |
| | EMS workload | 2 |
| | Chain relocation effects | 1 |

*Each study could include several variables.
EMS, emergency medical service.

positive, negative and neutral associations with trauma patient mortality.[25] Patients with critical conditions, such as cardiac arrests, are likely to benefit from also small improvements in response time. However, the actual time gained by using dynamic ambulance relocations, in comparison to static ambulance allocation (eg, the ordinary ambulance station), was only studied in three of the articles, and in those, the time saved rendered from a few seconds up to 2 min. Hence, whether this saved time had clinical relevance (ie, affected the outcome of the patients) cannot be determined as it was not measured. Dynamic ambulance relocation strategies focusing solely on shortening the average response time might be counter-productive when trying to save lives and minimise harm, especially in rural areas, since it may case longer response times or turnaround times for other patients.

The goal must be to reach as many patients as possible with time-critical conditions 'fast enough', not shortening the average response time or turnaround times. Reaching some patients fast enough might mean that other patients must wait longer for an ambulance. Less populated areas might have to rely on other rapid response units for the initial and potentially life-saving response before an ambulance can arrive. Since prehospital time is most likely not the only variable effecting clinical outcomes and quality of service within the ambulance services, variables such as early identification of time-critical conditions, prehospital medical interventions provided[24] and the actual way of driving the ambulance[26] needs to be taken into consideration when discussing the optimal ambulance service. Studies evaluating the effect of ambulance relocation with respect to patient outcome with respect to the global effect of the relocation, that is, not only on singular medical conditions but rather the effect on the patient population serviced by the relocated ambulances in real life contexts are strongly needed.

This review also does not allow any judgement of the value of dynamic relocation based on turnaround times, as only two of the included studies reported turnaround times. Both of these studies were conducted in large cities with many available emergency departments and neither compared turnaround times when dispatching using fixed ambulance locations versus dynamic relocation. Dynamic ambulance relocations may also cause a chain reaction, as temporary relocation of one ambulance may lead to a lack of resources in the original location, thus leading to new relocations and so on. These effects were covered in only one of the included studies. Therefore, no conclusions can be drawn regarding reductions in the response time or turnaround time by dynamic ambulance relocation based on the published studies.

Most studies developed and evaluated models based on EMS in urban areas. Notable differences exist between urban and rural areas regarding ambulance coverage, and turnaround time is most likely shorter in a city than in rural areas. A shortage of ambulance resources may, therefore, be more critical in rural areas than in urban areas. If the number of ambulances is small, the effect of a shortage or temporary relocation of one ambulance is more noticeable than in a city with more ambulances per area.[27] Conversely, the statistical risk for the occurrence of a time-critical condition is considerably lower in rural areas with small populations than in cities with large populations. However, the distances might be longer in rural areas than in urban areas. Even a temporary relocation may cause a significant delay in EMS response time; therefore, the effects of dynamic relocation might be more prominent in rural areas. Given the vulnerability of EMS coverage in rural areas, dynamic ambulance relocation in these areas needs to be prioritised in future studies. An important point made by Bélanger et al[18] is that dynamic ambulance relocation strategies could not compensate for a general lack of resources in a longer perspective and had side effects such as prolonged travelled distance

and related costs. This was also the only identified study reflecting on optimal ambulance use from an economical or environmental perspective.

The well-being and working environment of EMS personnel is another relevant aspect of the optimal use of available ambulances. Long working hours, shift work and short times between the missions used for recovery are associated with work-related stress and mental health problems.[28] A practical consequence of dynamic ambulance relocation can be limited access to a toilet, kitchen or resting facilities for ambulance personnel. Hence, human resource management may be more challenging when using dynamic dispatch strategies compared with fixed ambulance locations.[18] In this review, only one model included variables associated with the ambulance personnel working times, shifts and night rest. Only one study reported on workload effects. Working hours, working environment and workload are all important for both the effectiveness and efficiency of the EMS and should be considered when creating models for dynamic ambulance relocations.

## Limitations

This scoping review has several limitations. One was that the search was performed in three databases. However, these provided a broad selection of both medical and non-medical journals and were, therefore, considered, sufficient. By including studies from medical, technical and mathematical disciplines, the format for paper presentations varied. According to the scoping review methodology, only data from the results section should be charted and used in the review results. However, since the structure of the included papers varied, valuable information or outcomes stated under other headings were also extracted and included in the results of this review if the data or outcome referred to the actual study. The search strategy was revised several times during the revision process of this paper. The inconsistent terminology and definitions used for ambulance dispatching and dynamic relocation make it difficult to identify all available papers and motivated a free text search in addition to the systematic searches. A quality appraisal is not mandatory when using the scoping review methodology. However, to judge the value of the findings, a quality appraisal was found to add value and was therefore conducted.

## CONCLUSION

The current knowledge on dynamic relocation emanates from mathematical and technical support used to calculate the optimal locations of ambulance services. However, knowledge of how patient outcomes and working environments are affected by dynamic ambulance dispatching is lacking. This review has highlighted several gaps in the scientific coverage of the topic. The primary concerns are the lack of studies reporting on patient outcomes and the limited knowledge about several key factors, including the optimal use of ambulances in rural areas, turnaround

times, chain effects and aspects of a working environment for EMS personnel. Therefore, addressing these knowledge gaps will be important in future studies.

**Acknowledgements** The authors would like to thank academic librarians Ulrika Johansson (Örebro University library) and Lina Puhakka (Örebro University library) for their support in the literature search process.

**Contributors** JB: conceptualisation, data collection, analysis, writing of the manuscript; LK: conceptualisation, analysis, writing of the manuscript; EH: conceptualisation, data collection, confirming analysis, writing of the manuscript; KH: conceptualisation, data collection, analysis, writing and editing of the manuscript and revisions. All authors read and approved the final manuscript.

**Funding** The authors have not declared a specific grant for this research from any funding agency in the public, commercial or not-for-profit sectors.

**Competing interests** None declared.

**Patient and public involvement** Patients and/or the public were not involved in the design, or conduct, or reporting, or dissemination plans of this research.

**Patient consent for publication** Not applicable.

**Provenance and peer review** Not commissioned; externally peer reviewed.

**Data availability statement** Not applicable.

**ORCID iD**
Karin Hugelius http://orcid.org/0000-0003-0534-4593

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
