## [Reviewer comments · BMJ Open]

ARTICLE DETAILS

TITLE (PROVISIONAL)	Dynamic ambulance relocation: a scoping review
AUTHORS	Becker, Julia; Kurland, Lisa; Höglund, Erik; Hugelius, Karin

VERSION 1 – REVIEW

REVIEWER	Anderson, Melanie University Health Network, Library and Information Services
REVIEW RETURNED	17-Apr-2023

GENERAL COMMENTS	A scoping review is a good choice for this question, as it appears to be an emerging area of research with important gaps that should be addressed. Unfortunately there are several methodological errors that mean this review does not meet quality standards for a scoping review. The reporting of the search strategies is inadequate. Reporting of search strategies in a scoping review should be aiming for reproducibility - in other words a reader should be able to redo the reviewers' searches exactly as they did them. The table providing the details of the searches does not include information about what fields were used, what subject headings were used in PubMed if any, or how the "Totals" were arrived at based on the previous search lines. Web of Science is a platform that has different databases in it depending on the subscriptions of the institution, so for any Web of Science strategy to be reproducible the names of the databases that were included in the search need to be listed with the WoS strategy. More reproducible ways to provide the search strategies than a table in the body of the article would be to create an appendix with the full and complete search strategies directly from the databases included or to make such a document available in a data repository and cite it in this review. The search strategies are inadequate. No subject headings appear to be used in PubMed, no truncation or adjacencies are used in Scopus or Web of Science, and all searches rely almost entirely on one keyword per concept. It is good to see that the authors obtained a librarian's advice in creating the strategies, but these strategies would only be adequate as a basic search for background information and if submitted for PRESS peer review by another librarian they would not have passed without significant alteration. For a scoping review a search should be aiming to be as exhaustive as possible. This would mean using all of the synonyms that any researcher is likely to have used for each of the concepts the reviewers are looking for. In this case, especially since this appears to be an area that is emerging and most likely does not have hardened vocabulary, it would be important to brainstorm different words that authors might use in articles about this topic. It is unlikely that "dynamic" is going to be the only word that every author will use to describe a responsive or adaptive or
---

	data-driven dispatching system. It is also unlikely that the only word that would be used to describe the distribution or deployment or allocation or routing of ambulances would be "dispatch". Since the search that has been conducted for this review only uses these words, it is more accurate to say it is a review of articles *that use these words* than it is to say it is a review of articles that are *about this topic/question*. A not insignificant percentage of articles addressing these kinds of methods of ambulance distribution will be technology focused, it would also be advisable to consider including words like algorithm, "machine learning" etc. as part of the "dynamic" concept. The selection of the databases is adequate, although if the authors have access to Medline on another platform - Ovid or EBSCO in particular - they will have an easier time conducting the kind of search that would be appropriate for a scoping review. Many advanced search features are not available in PubMed, and PubMed's algorithm both makes searches less reproducible and sometimes interprets terms incorrectly. Depending on access, options to improve the coverage of this review would be to include Embase, which is an important medical database that covers research not found in Medline, WoS or Scopus, and consider discrete computer science databases if the development/evaluation of the technology would be of interest.
--	---

REVIEWER	Voss, Sarah University of the West of England, Health and Life Sciences
REVIEW RETURNED	21-Apr-2023

GENERAL COMMENTS	total of 18 papers were included in the review which highlights that there is a lack of evidence about the impact of the strategy on patient outcomes and staff well-being/working environment. This is a well-written paper on a topic that is interesting to a wide audience, including EMS researchers and those with a focus on service organisation and delivery. My overarching concern with the paper is related to the key finding. Dynamic ambulance location is a complicated area of research that has received considerable interest with a focus on improving response times for critically unwell patients. I am not entirely convinced that it would be plausible to design and conduct research that directly assesses the impact of dynamic ambulance relocation, rather than response times, on patient outcomes and I think this paper would benefit from a clear explanation of why this gap needs to be addressed and why response times (depending on call category) and turnaround times are not sufficient. In the discussion, the authors point out that trauma is a time-critical medical condition and question if differences of one minute are of clinical relevance. However, cardiac arrest patients, or critically unwell patients at imminent risk of cardiac arrest, are extremely likely to benefit from this time saving. There a few minor points that also need to be addressed: The searches were limited to 2012-2022 but 2013-2022 were the dates used in the selection criteria – there is a 2012 paper included so assume this is a typographical error. Were there any disagreements during the screening of papers and if so, how were these resolved? The patient and public involvement statement need to be revised to explain if/how patients or members of the public contributed to the research aims, design and reporting.
---

	The dates in the results state papers were included between 2014 and 2022 but I think this should be from 2012.
--	---

REVIEWER	Madathil, Sreenath Chalil Binghamton University
REVIEW RETURNED	28-Apr-2023

GENERAL COMMENTS	The authors concluded that Patient outcomes and working environments are lacking in the available literature. However several papers such as Enayati et al., and van Burren and others do include these variables as timely coverage and EMS workloads. A clarification can help to further this scoping review paper. Overall, the scoping review paper is well written
--

VERSION 1 – AUTHOR RESPONSE

Reviewer: 1	
Ms. Melanie Anderson, University Health Network Comments to the Author:	
A scoping review is a good choice for this question, as it appears to be an emerging area of research with important gaps that should be addressed. Unfortunately there are several methodological errors that mean this review does not meet quality standards for a scoping review. The reporting of the search strategies is inadequate. Reporting of search strategies in a scoping review should be aiming for reproducibility - in other words a reader should be able to redo the reviewers' searches exactly as they did them. The table providing the details of the searches does not include information about what fields were used, what subject headings were used in PubMed if any, or how the "Totals" were arrived at based on the previous search lines. Web of Science is a platform that has different databases in it depending on the subscriptions of the institution, so for any Web of Science strategy to be reproducible the names of the databases that were included in the search need to be listed with the WoS strategy. More reproducible ways to provide the search strategies than a table in the body of the article would be to create an appendix with the full and complete search strategies directly from the databases included or to make such a document available in a data repository and cite it in this review.	Thank you for providing important and supportive suggestions to our scoping review. We appreciate your time and concern and have made a new, updated search with new terms (both MeSH and new keywords) and added synonyms to the previous search. The previous Table 1 has been replaced with a supplementary file with the search strategy for all three databases, providing information on database, terms and fields used. We have used truncation and Boolean operators to support the search.

The search strategies are inadequate. No subject headings appear to be used in PubMed, no truncation or adjacencies are used in Scopus or Web of Science, and all searches rely almost entirely on one keyword per concept. It is good to see that the authors obtained a librarian's advice in creating the strategies, but these strategies would only be adequate as a basic search for background information and if submitted for PRESS peer review by another librarian they would not have passed without significant alteration. For a scoping review a search should be aiming to be as exhaustive as possible. This would mean using all of the synonyms that any researcher is likely to have used for each of the concepts the reviewers are looking for. In this case, especially since this appears to be an area that is emerging and most likely does not have hardened vocabulary, it would be important to brainstorm different words that authors might use in articles about this topic. It is unlikely that "dynamic" is going to be the only word that every author will use to describe a responsive or adaptive or data-driven dispatching system. It is also unlikely that the only word that would be used to describe the distribution or deployment or allocation or routing of ambulances would be "dispatch". Since the search that has been conducted for this review only uses these words, it is more accurate to say it is a review of articles *that use these words* than it is to say it is a review of articles that are *about this topic/question*.	
A not insignificant percentage of articles addressing these kinds of methods of ambulance distribution will be technology focused, it would also be advisable to consider including words like algorithm, "machine learning" etc. as part of the "dynamic" concept.	Thank you for this wise suggestion. Both the keyword dynamic allocation and the MesH term machine learning have been added to the search and a new search has been conducted.
The selection of the databases is adequate, although if the authors have access to Medline on another platform - Ovid or EBSCO in particular - they will have an easier time conducting the kind of search that would be appropriate for a scoping review. Many advanced search features are not available in PubMed, and PubMed's algorithm both makes searches less reproducible and sometimes interprets terms incorrectly. Depending on access, options to improve the coverage of this review would be to include Embase, which is an	Thank you for this comment. In accordance with the advice from our academic librarian and previous scoping reviews published in BJI Open, we have stayed with the PubMed search for this paper, since the search is very broad and specific MesH terms are lacking within this topic. We also would have liked to choose using Embase, but since we do not have institutional access to this database and have made a structured search in Scopus, that also covers

important medical database that covers research not found in Medline, WoS or Scopus, and consider discrete computer science databases if the development/evaluation of the technology would be of interest.	most of the content of the Embase and several more sources, we decided to stay with the Scopus.
Reviewer: 2 Dr. Sarah Voss, University of the West of England Comments to the Author:	
A total of 18 papers were included in the review which highlights that there is a lack of evidence about the impact of the strategy on patient outcomes and staff well-being/working environment. This is a well-written paper on a topic that is interesting to a wide audience, including EMS researchers and those with a focus on service organisation and delivery.	Thank you for this encouraging comment. We agree that this is an area that deserves more attention and hope that the current manuscript will contribute.
My overarching concern with the paper is related to the key finding. Dynamic ambulance location is a complicated area of research that has received considerable interest with a focus on improving response times for critically unwell patients. I am not entirely convinced that it would be plausible to design and conduct research that directly assesses the impact of dynamic ambulance relocation, rather than response times, on patient outcomes and I think this paper would benefit from a clear explanation of why this gap needs to be addressed and why response times (depending on call category) and turnaround times are not sufficient. In the discussion, the authors point out that trauma is a time-critical medical condition and question if differences of one minute are of clinical relevance. However, cardiac arrest patients, or critically unwell patients at immanent risk of cardiac arrest, are extremely likely to benefit from this time saving.	Thank you for this wise comment. We agree that this is a complicated and complex question. We have added some text on this question and why we think more studies are needed that explore medical outcomes. We also agree with you that some patients, especially in urban settings, are likely to benefit from also small improvements of response time and have revised this finding.
There a few minor points that also need to be addressed: The searches were limited to 2012-2022 but 2013-2022 were the dates used in the selection criteria – there is a 2012 paper included so assume this is a typographical error.	Thank you for notifying us in this matter, we have revised this throughout the manuscript.
Were there any disagreements during the screening of papers and if so, how were these resolved?	Thank you for adding this important question. We have added the following text in the selection section to address this: One disagreement occurred during the selection process and it was solved by involving the second author (LK) in the decision.

The patient and public involvement statement need to be revised to explain if/how patients or members of the public contributed to the research aims, design and reporting.	Thank you for this comment. We agree that this statement seems a bit short, but have followed the editorial office instructions on how they want this statement to be formulated.
The dates in the results state papers were included between 2014 and 2022 but I think this should be from 2012.	Thank you for notifying us on this, it has been corrected throughout the manuscript.
Reviewer: 3 Dr. Sreenath Chalil Madathil, Binghamton University Comments to the Author:	
The authors concluded that Patient outcomes and working environments are lacking in the available literature. However several papers such as Enayati et al., and van Burren and others do include these variables as timely coverage and EMS workloads. A clarification can help to further this scoping review paper. Overall, the scoping review paper is well written.	Thank you for this comment. We agree that this needs to be revised and have revised in accordance and highlighted that some papers actually refers to a more complex perspective of these questions.

VERSION 2 – REVIEW

REVIEWER	Anderson, Melanie University Health Network, Library and Information Services
REVIEW RETURNED	28-Aug-2023

GENERAL COMMENTS	Thank you for your responses and the improvements to the search and documentation. Unfortunately the search continues to rely on an inadequate selection of terms. In PubMed the section for ambulance contains no keywords at all and relies entirely on MeSH, therefore missing all of the unindexed or poorly indexed contents of PubMed. The MeSH Emergency Medical Dispatch is not used. The section for "dynamic" is missing terms like artificial intelligence, the keyword for machine learning, algorithm, data-driven, data-based, responsive, etc. It continues to be important that articles not be accidentally excluded because their authors chose different words to describe something that would count as "dynamic ambulance dispatch".
---

REVIEWER	Voss, Sarah University of the West of England, Health and Life Sciences
REVIEW RETURNED	04-Sep-2023

GENERAL COMMENTS	I am satisfied that my original comments have been addressed
--

VERSION 2 – AUTHOR RESPONSE

Reviewer: 1 Ms. Melanie Anderson, University Health Network Comments to the Author:

Thank you for your responses and the improvements to the search and documentation. Unfortunately the search continues to rely on an inadequate selection of terms. In PubMed the section for ambulance contains no keywords at all and relies entirely on MeSH, therefore missing all of the unindexed or poorly indexed contents of PubMed. The MeSH Emergency Medical Dispatch is not used. The section for "dynamic" is missing terms like artificial intelligence, the keyword for machine learning, algorithm, data-driven, data-based, responsive, etc. It continues to be important that articles not be accidentally excluded because their authors chose different words to describe something that would count as "dynamic ambulance dispatch".

Thank you again for providing important and supportive suggestions regarding the search strategy. We appreciate your time and concern and have made a new, updated search in the databases and a new inclusion- and exclusion study selection based on the new search.

Both more MESH terms and keywords were included in our new search. We recognize that unindexed or poorly indexed publications can be difficult to identify using eg MESH terms, which is why search in title and/ or abstract was performed.

The supplementary file describing the literature search has been updated and attached.

Reviewer 2. Prof. Sarah Voss, University of the West of England

I am satisfied that my original comments have been addressed.

Thank you for your time and efforts to improve our paper.